# A Novel Variational Family for Hidden Non-linear Markov Models

## Abstract

Latent variable models have been widely applied for the analysis and visualization of large datasets. In the case of sequential data, closed-form inference is possible when the transition and observation functions are linear. However, approximate inference techniques are usually necessary when dealing with nonlinear evolution and observations. Here, we propose a novel variational inference framework for the explicit modeling of time series, Variational Inference for Nonlinear Dynamics (VIND), that is able to uncover nonlinear observation and latent dynamics from sequential data. The framework includes a structured approximate posterior, and an algorithm that relies on the fixed-point iteration method to find the best estimate for latent trajectories. We apply the method to several datasets and show that it is able to accurately infer the underlying dynamics of these systems, in some cases substantially outperforming state-of-the-art methods.

## 1 Introduction

In recent years, advances on neural data acquisition have made it possible to record the simultaneous sequential activity of up to thousands of neurons (Paninski & Cunningham (2017)). The analysis of these datasets often focuses on dimensionality reduction techniques that encode the activity of the population in a lower dimensional latent trajectory (Cunningham & Yu (2014)). At the other extreme, there is a big body of detailed electrophysiological data coming from voltage measurements in single cells (Jones et al. (2009)). In this setting it is understood that the underlying dynamics is in fact highly nonlinear and multidimensional, though the experimenter only has access to a one-dimensional (1D) observation. From such 1D recordings, the task is to approximately recover the complete hidden latent space paths and dynamics.

A host of sophisticated techniques has been proposed for the analysis of complex sequential data that is not well described by linear transitions and observations (Archer et al. (2015); Chung et al. (2015); Gao et al. (2016; 2015); Hernandez et al. (2017); Johnson et al. (2016); Krishnan et al. (2015; 2016); Linderman et al. (2017); Pandarinath et al. (2018); Sussillo et al. (2016); Zhao & Memming Park (2017); Wu et al. (2017; 2018)). In that context, we here present Variational Inference for Nonlinear Dynamics (VIND). The main contribution of VIND is an algorithm that allows variational inference (VI) from structured, intractable approximations to the posterior distribution. In particular, VIND can handle variational posteriors that (i) represent nonlinear evolution in the latent space, and (ii) disentangle the latent dynamics (transition) from the data encoding (recognition). Crucially, the VIND approximate posterior shares the exact nonlinear structure of latent dynamics evolution with the model for data generation. This makes the VIND approximation potentially more powerful than models in which the choice of approximate posterior is made solely on grounds of tractability.

VIND relies on two key ideas. Firstly, it makes use of the fact that given an intractable posterior $Q(\mathbf{Z}|\mathbf{X})$, it is always possible to compute a tractable Gaussian approximation to it. This Gaussian approximation inherits its parameters from $Q(\mathbf{Z}|\mathbf{X})$ (Chung et al. (2015)), so optimizing for it can be interpreted as indirectly optimizing $Q(\mathbf{Z}|\mathbf{X})$. The second novel aspect of VIND is the use of the fixed-point iteration (FPI) method to significantly speed up the computation of the aforementioned Gaussian approximation.

In this work we focus on a VIND variant in which the latent dynamics is represented as a Locally Linear Dynamical System (LLDS). The running time of LLDS/VIND is linear in the number of time points in a trial. We are especially interested in determining LLDS/VIND's ability to infer the

hidden dynamics, as demonstrated by its generative / predictive capabilities. After training, can the VIND-trained model generate data that is indistinguishable from the original observations, if provided with a suitable starting point? In the second half of this work we apply VIND to four datasets, one synthetic and three using experimental data, and show that VIND excels in this task, in some cases outperforming established methods by orders of magnitude in the predictive mean squared error (MSE).

## 2 VARIATIONAL INFERENCE WITH NONLINEAR DYNAMICS (VIND)

For a set of temporally ordered, correlated, noisy observations $\mathbf{X} \equiv \{\mathbf{x}_1, \ldots \mathbf{x}_T\}$, $\mathbf{x}_t \in \mathbb{R}^{d_X}$, a latent variable model proposes an additional, time-ordered set of random variables $\mathbf{Z} \equiv \{\mathbf{z}_1, \ldots \mathbf{z}_T\}$, $\mathbf{z}_T \in \mathbb{R}^{d_Z}$ that is hidden from view. The hidden state $\mathbf{z}_t$ is endowed with a stochastic dynamics: $\mathbf{z}_{t+1} \sim p(\mathbf{z}_{t+1}|\mathbf{z}_{1:t})$ by which it evolves. The observations $\mathbf{x}_t$ are generated by drawing samples from a $\mathbf{z}_t$-dependent probability distribution.

A naive objective for such a model is the marginal log-likelihood $\log p(\mathbf{X})$, with the latent variables integrated out of the joint. However, it is well known that for anything other than the simplest distributions, marginalization with respect to $\mathbf{Z}$ is intractable (Bishop (2006)). VI overcomes this problem by approximating the posterior $p(\mathbf{Z}|\mathbf{X})$ with a distribution $q(\mathbf{Z}|\mathbf{X})$, the Recognition Model (RM), from a tractable class. The objective becomes the celebrated ELBO, a lower bound to $\log p(\mathbf{X})$ (Jordan et al. (1999)):

$$\log p(\mathbf{X}) \geq \mathscr{L}_{\text{ELBO}}(\mathbf{X}) = \mathbb{E}_q[\log p(\mathbf{X}, \mathbf{Z})] - \mathbb{E}_q[\log q(\mathbf{Z}|\mathbf{X})]. \tag{1}$$

Successful VI relies on the choice of the approximation $q(\mathbf{Z}|\mathbf{X})$. This choice is constrained by two goals that stand in tension: expressiveness and tractability. We approach the search for a suitable class of variational posterior by considering the joint $p(\mathbf{X}, \mathbf{Z})$. Our focus is on factorizations of the form:

$$p(\mathbf{X}, \mathbf{Z}) \equiv p_{\phi,\theta}(\mathbf{X}, \mathbf{Z}) = c_{\phi,\theta} \cdot H_\phi(\mathbf{Z}) \prod_{t=0}^{T} f_\theta(\mathbf{x}_t|\mathbf{z}_t), \tag{2}$$

where the distribution parameters have been written explicitly. The unnormalized densities $f_\theta$ stand for an observation model that, for the purposes of this work, can be either Gaussian, $\mathbf{x}_t|\mathbf{z}_t \sim \mathcal{N}(m_\theta(\mathbf{z}_t), \boldsymbol{\Sigma})$, or Poisson, $\mathbf{x}_t|\mathbf{z}_t \sim \text{Poisson}(\lambda_\theta(\mathbf{z}_t))$. The respective mean, $m_\theta(\mathbf{z}_t)$, and rate, $\lambda_\theta(\mathbf{z}_t)$, are arbitrary nonlinear functions of the latent state $\mathbf{z}_t$, that we represent as neural networks. The standard deviation $\boldsymbol{\Sigma}$ of the Gaussian observation model is taken to be $\mathbf{z}_t$-independent. $c_{\phi,\theta}$ is a normalization constant. $H_\phi$ is the latent evolution term in $\mathbf{Z}$-space with a Markov Chain structure Johnson et al. (2016); Krishnan et al. (2015; 2016); Archer et al. (2015); Gao et al. (2016):

$$H_\phi(\mathbf{Z}) = h_0(\mathbf{z}_0) \prod_{t=1}^{T} h_\phi(\mathbf{z}_t|\mathbf{z}_{t-1}), \tag{3}$$

$$\mathbf{z}_0 \sim \mathcal{N}(a_0, \Gamma_0), \tag{4}$$

$$\mathbf{z}_t|\mathbf{z}_{t-1} \sim \mathcal{N}(a_\phi(\mathbf{z}_{t-1}), \Gamma), \tag{5}$$

where $a_\phi(\mathbf{z})$ is an arbitrary nonlinearity.

From Eq. (2), the posterior distribution of the Generative Model (GM) can be factorized as

$$p_{\phi,\theta}(\mathbf{Z}|\mathbf{X}) = \frac{c_{\phi,\theta} \prod f_\theta(\mathbf{x}_t|\mathbf{z}_t)}{p_{\phi,\theta}(\mathbf{X})} \cdot H_\phi(\mathbf{Z}). \tag{6}$$

We used Eq. (6) as a guidance to propose the inclusion of the GM evolution term $H_\phi(\mathbf{Z})$ into the variational posterior. We start then with an approximation that factorizes as:

$$Q_{\phi,\varphi}(\mathbf{Z}|\mathbf{X}) = \alpha_{\phi,\varphi}(\mathbf{X}) \, G_\varphi(\mathbf{X}, \mathbf{Z}) H_\phi(\mathbf{Z}). \tag{7}$$

We refer to $Q_{\phi,\varphi}(\mathbf{Z}|\mathbf{X})$ as the *parent* distribution. By design, the factor $G_\varphi$ in the parent contains all the dependence on $\mathbf{X}$. For definiteness, the case

$$G_\varphi(\mathbf{X}, \mathbf{Z}) = \prod_{t=0}^{T} g_\varphi(\mathbf{z}_t|\mathbf{x}_t), \quad \mathbf{z}_t|\mathbf{x}_t \sim \mathcal{N}(\boldsymbol{\mu}_\varphi(\mathbf{x}_t), \boldsymbol{\Lambda}_\varphi^{-1}(\mathbf{x}_t)), \tag{8}$$

is considered in this work, where $\boldsymbol{\mu}(\mathbf{x})$ and $\boldsymbol{\Lambda}(\mathbf{x})$ are nonlinear maps.

In Eq. (7), $\alpha_{\phi,\varphi}$ is a normalization constant that, regardless of the specific form of $G_\varphi$, cannot be computed in closed form. This is most easily seen by noticing that due to the nonlinearity $h(\mathbf{z}_T|\mathbf{z}_{T-1})$, integration with respect to $\mathbf{z}_T$ yields an intractable $\mathbf{z}_{T-1}$-dependent factor. Hence, VI cannot be formulated directly in terms of the parent, since expectation values with respect to $Q_{\phi,\varphi}$ cannot be computed in closed form.

VIND represents a way out of this conundrum that, effectively, allows for the use of an intractable, unnormalized distribution as the Recognition Model in VI. The idea is to define a Gaussian approximation $q_{\phi,\varphi}(\mathbf{Z}|\mathbf{X})$ to the parent; this *child* distribution is then used as the variational posterior in Eq. (1). The inference problem becomes tractable since the child is normal. Importantly, the parameters in $q_{\phi,\varphi}(\mathbf{Z}|\mathbf{X})$, with respect to which we optimize, are inherited from the parent. After training, they can be replaced back into $Q_{\phi,\varphi}(\mathbf{Z}|\mathbf{X})$ obtaining, in particular, the nonlinear dynamics $a_\phi(\mathbf{z})$ for the latent space. In this novel way, VIND achieves the *reuse* of the latent dynamics $h(\mathbf{z}_t|\mathbf{z}_{t-1})$ of the GM in the RM.

Concretely, let $q_{\phi,\varphi}$ be a Laplace approximation to $Q_{\phi,\varphi}$,

$$q_{\phi,\varphi}(\mathbf{Z}|\mathbf{X}) = \mathcal{N}\big(\mathbf{P}_{\phi,\varphi}(\mathbf{X}), \mathbf{C}_{\phi,\varphi}^{-1}(\mathbf{X})\big) . \tag{9}$$

The mean $\mathbf{P}_{\phi,\varphi}$ in Eq. (9) is the solution to the equation

$$\frac{\partial}{\partial \mathbf{Z}} \log Q_{\phi,\varphi}(\mathbf{Z}|\mathbf{X})\bigg|_{\mathbf{Z}=\mathbf{P}} = \mathbf{0} , \tag{10}$$

and the precision is given by

$$[\mathbf{C}_{\phi,\varphi}(\mathbf{X})]_{ij} = \frac{\partial^2}{\partial \mathbf{Z}_i \partial \mathbf{Z}_j} \log Q_{\phi,\varphi}(\mathbf{Z}|\mathbf{X})\bigg|_{\mathbf{Z}=\mathbf{P}_{\phi,\varphi}(\mathbf{X})} \equiv \big[s_{\phi,\varphi}\big(\mathbf{P}_{\phi,\varphi}(\mathbf{X}), \mathbf{X}\big)\big]_{ij}, \tag{11}$$

where Eq. (11) defines $s_{\phi,\varphi}$. A closed form solution for Eq. (10) is not possible in general. However, for a large class of distributions, and in particular for any $Q_{\phi,\varphi}$ such that $\log Q_{\phi,\varphi}$ includes terms quadratic in $\mathbf{Z}$, it is possible to rewrite Eq. (10) in the form

$$\mathbf{P} = r_{\phi,\varphi}(\mathbf{P}, \mathbf{X}) . \tag{12}$$

In this form, the latter can be approximately solved by making use of the FPI method. This approximate solution provides the mean for the variational approximation $q_{\phi,\varphi}(\mathbf{Z}|\mathbf{X})$.

Henceforth, VIND's algorithm includes two steps per epoch that are carried out in alternation (see Algorithm 1 in App. A). The first step is a FPI that, for the current values of the parameters $\phi, \varphi$, determines the mean and variance of a Laplace approximation to the parent. The second is a regular ADAM gradient ascent update Kingma & Ba (2014) with respect to the ELBO objective. Upon convergence, the set of parameters $\phi, \varphi, \theta$ that maximizes the ELBO can be plugged into $a_\phi(\mathbf{z})$, to obtain a dynamical rule that interpolates between the different latent trajectories inferred from the data trials.

In the experiments conducted in this paper, the nonlinear dynamics is specified as $a_\phi(\mathbf{z}) = A_\phi(\mathbf{z})\mathbf{z}$ where $A_\phi(\mathbf{z})$ is a state-space dependent $d_Z \times d_Z$ matrix. We call this evolution rule, a Locally Linear Dynamical System, and the resulting inference algorithm LLDS/VIND. To derive it, consider a parent distribution distribution $Q_{\phi,\varphi}$, as defined in Eq. (7). The mean $\boldsymbol{\mu}_\varphi$ and the standard deviation $\boldsymbol{\Lambda}_\varphi^{-1}$ in Eq. (8) are represented as deep neural networks:

$$\boldsymbol{\mu}_\varphi = \mathrm{NN}_{\varphi_{\boldsymbol{\mu}}}(\mathbf{x}_t), \quad \boldsymbol{\Lambda}_\varphi = \mathrm{NN}_{\varphi_{\boldsymbol{\Lambda}}}(\mathbf{x}_t) . \tag{13}$$

The remaining ingredient of $Q_{\phi,\varphi}$ is the shared evolution law $H_\phi$, Eq. (3). We write the $h_\phi$ factors that determine the latent evolution model as

$$h_\varphi(\mathbf{z}_{t+1}|\mathbf{z}_t) = \exp\left\{-\frac{1}{2}\Big(\mathbf{z}_{t+1} - A_\varphi(\mathbf{z}_t)\mathbf{z}_t\Big)^T \Gamma\Big(\mathbf{z}_{t+1} - A_\varphi(\mathbf{z}_t)\mathbf{z}_t\Big)\right\} , \tag{14}$$

where $\Gamma$ is a constant precision matrix. Eq. (14) can be thought of as describing the stochastic evolution of a "locally linear" dynamical system:

$$\mathbf{z}_{t+1} \sim A(\mathbf{z}_t)\mathbf{z}_t + \text{noise} . \tag{15}$$

By LLDS/VIND we refer to VIND with this specific parameterization of the nonlinearity $a_\phi(\mathbf{z}_t)$ in Eq. (5). LLDS/VIND has some desirable features:

1. The limit of linear evolution is easily taken as $A_\phi(\mathbf{z}_t) \to$ const..
2. $\max_{\mathbf{Z}} |A_\phi(\mathbf{z}_t) - \mathbb{I}|$ is a simple measure of the smoothness of the latent trajectories.

The equation for the Laplace mean is found by taking derivatives of $Q_{\phi,\varphi}$ with respect to the hidden path:

$$\left. \frac{\partial \log Q_{\phi,\varphi}}{\partial \mathbf{Z}} \right|_{\mathbf{Z}=\mathbf{P}} = 0 \,. \tag{16}$$

Using Eqs. (3) and (8) we find

$$\log Q_{\phi,\varphi} = \log C_{\phi,\varphi} - \frac{1}{2} \left[ (\mathbf{Z} - \mathbf{M}_\varphi)^T \mathbf{\Lambda}_\varphi (\mathbf{Z} - \mathbf{M}_\varphi) + \mathbf{Z}^T \mathbf{S}_\phi(\mathbf{Z}) \mathbf{Z} \right] \tag{17}$$

where $\mathbf{M}_\varphi = \{\boldsymbol{\mu}_\varphi(\mathbf{x}_1), \ldots, \boldsymbol{\mu}_\varphi(\mathbf{x}_T)\}$ and $\mathbf{S}_\phi(\mathbf{Z})$ is a state-space-dependent covariance with tridiagonal form:

$$\mathbf{S}_\phi(\mathbf{Z}) = \begin{pmatrix} A_0^T \Gamma A_0 & -A_0^T \Gamma & 0 & \ldots & \ldots & 0 & 0 \\ -\Gamma A_0 & A_1^T \Gamma A_1 & A_1^T \Gamma & \ldots & \ldots & 0 & 0 \\ 0 & -\Gamma A_1 & \ldots & \ldots & \ldots & 0 & 0 \\ \ldots & \ldots & \ldots & \ldots & \ldots & \ldots & \ldots \\ 0 & 0 & \ldots & \ldots & \ldots & -A_{T-2}^T \Gamma & 0 \\ 0 & 0 & \ldots & \ldots & -\Gamma A_{T-2} & A_{T-1}^T \Gamma A_{T-1} & -A_{T-1}^T \Gamma \\ 0 & 0 & \ldots & \ldots & 0 & -\Gamma A_{T-1} & A_T^T \Gamma A_T \end{pmatrix} \tag{18}$$

Here $A_i \equiv A_\phi(\mathbf{z}_i)$.

Inserting Eq. (17) into Eq. (16) we obtain the LLDS/VIND equation for the posterior, Eq. (12), with

$$r_{\phi,\varphi}(\mathbf{P}, \mathbf{X}) = \left[ \mathbf{\Lambda}_\varphi + \mathbf{S}_\phi(\mathbf{P}) \right]^{-1} \left( \mathbf{\Lambda}_\varphi \mathbf{M}_\varphi - \frac{1}{2} \mathbf{P}^T \frac{\partial \mathbf{S}_\phi(\mathbf{P})}{\partial \mathbf{P}} \mathbf{P} \right) \,. \tag{19}$$

Note that it is not necessary to find the value of $C_{\phi,\varphi}$. Moreover, we remark that the matrix $\mathbf{\Lambda}_\varphi + \mathbf{S}_\phi(\mathbf{P})$ is block-tridiagonal and can be inverted in $O(T)$. Although Eq. (12) cannot be solved analytically, a solution can be approximated by choosing an initial point $\mathbf{P}^{(0)}$ and iterating

$$\mathbf{P}^{(n)} = r_{\phi,\varphi}(\mathbf{P}^{(n-1)}, \mathbf{X}) \,, \tag{20}$$

The VIND method assumes that this FPI converges. In practice, this assumption can be guaranteed throughout training by appropriate choices of hyperparameters and network architectures (see App. A.2).

During training, the mean $\mathbf{P}$ represents the best current estimate of the latent trajectory. Note that the FPI step in Eq. (20) mixes all the components in $\mathbf{P}$. In particular, the $i$-th component of $\mathbf{P}^{(n)}$ depends in general on all the components of $\mathbf{P}^{(n-1)}$ via the inverse covariance in Eq. (19). Thus, at every training epoch, the best estimate for the path at a specific time point $t$ contains information from the complete data, both to the past and to the future of $t$. This is analogous to the Kalman smoother.

## 3 RELATION TO PREVIOUS WORK

The problem of inference for sequential data has been treated extensively in the literature. The GfLDS and PfLDS models introduced in Archer et al. (2015); Gao et al. (2016) are particular cases of VIND in which the dynamics in the latent space is linear and time-invariant, i.e. $\mathbf{z}_t|\mathbf{z}_{t-1} \sim \mathcal{N}(A\mathbf{z}_{t-1}, \mathbf{Q})$. In the jargon used in this paper, this corresponds to the situation in which the parent distribution is Gaussian, and therefore equal to its own Laplace approximation. Eq. equation 10 can be solved analytically in this case and no FPI step is needed. Gaussian Process Factor Analysis (GPFA) Yu et al. (2009) assumes linear, time-invariant dynamics as well as a linear observation model, i.e. $\mathbf{x}_t|\mathbf{z}_t \sim \mathcal{N}(C\mathbf{z}_t + d, \mathbf{R})$, for some $C$, $d$, and $R$. We will be explicitly comparing our results to results obtained by these models.

In Krishnan et al. (2015), Deep Kalman Filters (DKF) were proposed to handle variational posterior distributions that describes nonlinear evolution in the latent space. Their approximate posterior, analogous to the parent distribution in this paper, is plugged directly into the ELBO. This imposes

some restrictions in the form the posterior can take - for instance, it must be Gaussian conditioned on the observations. VIND can handle factorizations of the parent distribution that are not restricted in this way, an example being LLDS/VIND, which has the form in Eq. equation 7. VIND's ability to handle unnormalizable parent distributions is due to the fact that VIND's actual approximate posterior is always strictly normal. The same authors built upon their idea in Krishnan et al. (2016), where a variational posterior was proposed that partially uses the conditional structure implied by the generative model. In this paper, a similar prescription is used by assuming that $Q_{\phi,\varphi}$ and $p_{\phi,\varphi}$ share exactly the same factorization for the latent evolution.

The authors of Johnson et al. (2016) combine probabilistic graphical models with message passing in an approach based on conjugate priors. The approximate posterior distributions considered in that work are restricted by the conjugacy requirements, in particular, the evolution term must belong to the exponential family. VIND's parent distribution is not subject to this requirement. However, since VIND's actual approximate posterior is still Gaussian, it may be possible to combine the two methods into one that can handle both nonlinear evolution and discrete latent variables.

In Chung et al. (2015), Gaussian noise is added to the deterministic evolution rule of an RNN in the context of a variational autoencoder, termed VRNN. Similarly to LLDS/VIND, these authors share the evolution factorization between the generative model and the approximate posterior and, indeed, the only difference between the structure of their model and that of LLDS/VIND is that the evolution there is expressed as an RNN instead of as an LLDS. However, their inference algorithm only uses past data to estimate the hidden state at any given time. VIND's algorithm, based on the FPI, uses information both from the past and from the future to estimate the latent paths. In Kalantari et al. (2018) a non-parametric approach was taken to determine the best latent dimension in an LDS. It would be interesting to apply those same methods to VIND. Finally, in Pandarinath et al. (2018); Sussillo et al. (2016) a sophisticated, bidirectional, Deep Learning-based RNN architecture called LFADS was proposed with neuroscience applications in mind. For both LFADS and DKF, we found difficult to modify their code to compute the quantities that are used in this paper to evaluate the quality of training. However, given the expressive power of these works, we expect them to perform comparably to VIND in the tasks considered in the next section.

## 4 RESULTS

We demonstrate the capabilities of LLDS/VIND by applying it to four characteristic datasets, each of which illustrates a VIND feature. First, we use synthetically generated 10D data with added Gaussian noise, and latent evolution determined by an Euler discretization of the Lorenz system. This dataset is the simplest and cleanly illustrates VIND's ability to infer the underlying nonlinear dynamics. Secondly we use a multi-electrode neural recording from a mouse performing a delayed-discrimination task. LLDS/VIND is run with both Gaussian and Poisson observation models. It is found that while a Gaussian observation model is superior for the explaining the variance in the data, the Poisson model performs better when it comes to interpolation of the dynamics. This is a common VIND tradeoff. The third dataset consists of a 1D voltage measurement from single-cell recordings. The problem in this case is not dimensionality reduction but rather to determine the underlying dynamics. We find that the minimum number of latent dimensions VIND requires to describe this dataset coincides with the naive expectation. Finally, we apply VIND to discovering dynamics in a more complex dataset coming from dorsal cortex calcium imaging, and find that it is possible to model the data using a surprisingly low number of latent dimensions. We also use VIND to reconstruct dynamics from one side of the brain to the other.

Given an inferred starting point in state-space, the quality of the dynamics uncovered by LLDS/VIND can be ascertained by evolving the system $k$ steps into the future *without* any input data. To clarify terminology, this is not strict prediction in the sense of pure extrapolation, since we use information about all $\mathbf{x}_t$, both in the past and in the future, to infer the starting point. In order to avoid doubt, we use the term *forward interpolate*. Forward interpolation essentially tests the extent to which the dynamics are accurately learned. We take VIND's capability for forward interpolation as the main measure of the fit's success. As we will show, this task remains highly challenging for simpler smoothing priors like the latent LDS, and it is one of the key strengths of VIND.

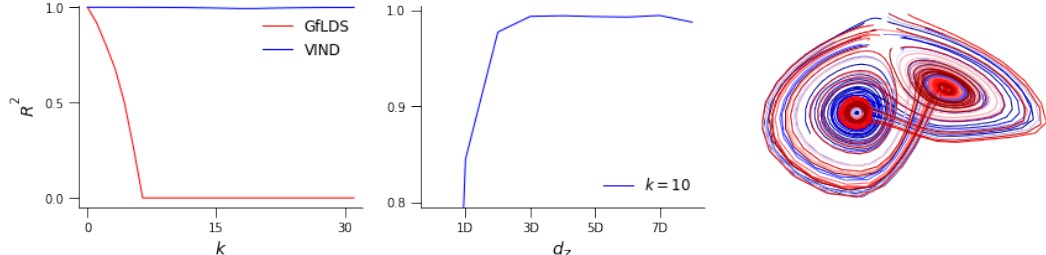

Figure 1: Comparison of results for the Lorenz dataset ($d_z = 3$) between GfLDS and VIND: (left) $R_k^2$ comparison; (center) $R_{10}^2$ as a function of dimension of the latent space; (right) VIND's inferred validation trajectories for this dataset.

To make this analysis quantitative, we compute the $k$-step mean squared error ($\text{MSE}_k$) on test data, and its normalized version, the $R_k^2$, defined as

$$\text{MSE}_k = \sum_{t=0}^{T-k} \left(\mathbf{x}_{t+k} - \hat{\mathbf{x}}_{t+k}\right)^2 , \quad R_k^2 = 1 - \frac{\text{MSE}_k}{\sum_{t=0}^{T-k} \left(\mathbf{x}_{t+k} - \bar{\mathbf{x}}\right)^2} \tag{21}$$

where $\bar{\mathbf{x}}$ is the data average for this trial and $\hat{\mathbf{x}}_{t+k}$ is the prediction at time $t+k$. The latter is obtained by i) using the full data $\mathbf{X}$ to obtain the best estimate for $\mathbf{z}_t$, ii) using $k$ times the LLDS/VIND evolution equation $\mathbf{z}_{t+1} = A_\varphi(\mathbf{z})\mathbf{z}_t$, or $\mathbf{z}_{t+1} = A\mathbf{z}_t$ for the LDSs, to find the latent state $k$ time steps in the future, and iii) using the generative network to compute the forward-interpolated observation. Note that in particular, $k = 0$ corresponds to the standard $R^2$. The more general $R_k^2$ ensures that VIND yields more than just a good autoencoder. We will be comparing results obtained with LLDS/VIND to several models, namely, GfLDS, PfLDS, and GPFA (see Sec. 3 for details).

### 4.1 LORENZ SYSTEM

The Lorenz system is a classical nonlinear differential equation in 3 independent variables.

$$\begin{aligned}
\dot{z}_1 &= \sigma(z_2 - z_1), \\
\dot{z}_2 &= z_1(\rho - z_3) - z_2, \\
\dot{z}_3 &= z_1 z_2 - \beta z_3.
\end{aligned} \tag{22}$$

This is a well studied system with chaotic solutions and that serves to cleanly demonstrate VIND's capabilities for inferring nonlinear dynamics. We generated numerical solutions of the Lorenz system from randomly generated initial conditions, for $\sigma = 10$, $\rho = 28$, $\beta = 8/3$, and additive Gaussian noise. Gaussian observations in a 10D space were then generated with the mean specified by a $\mathbf{z}$-dependent neural network. The complete synthetic data consisted of 100 trials, each comprising 250 time-steps, of which 80% was used for training and the remaining for validation.

The results of the fit to this data are shown in Fig. 1. The left panel shows the $R_k^2$ comparison for VIND and GfLDS fits, with $d_Z = 3$. Strikingly, for this dataset, VIND's performance is 2-3 orders of magnitude better than GfLDS. The right panel illustrates VIND's capability to infer properties of the underlying dynamics: VIND hits peak performance at $d_Z = 3$, the true dimensionality of this system. In the rightmost panel, all the paths inferred by VIND have been put together, showing the famous butterfly pattern.

### 4.2 ELECTROPHYSIOLOGY

We analyzed data collected from mice performing a delayed discrimination task in a simultaneous recording session, using multi-unit electrophysiology (64 channel Janelia silicon probe) Guo et al. (2014); Li et al. (2015). In this task, the animals were trained to discriminate the location of a pole using whiskers. The pole was presented at $t = -1.3$s, and an auditory go cue at $t = 0$ signaled the beginning of the response epoch. During response, the mice reported the perceived pole position by licking one of two lick ports. Neurons in this task exhibit complex dynamics across behavioral

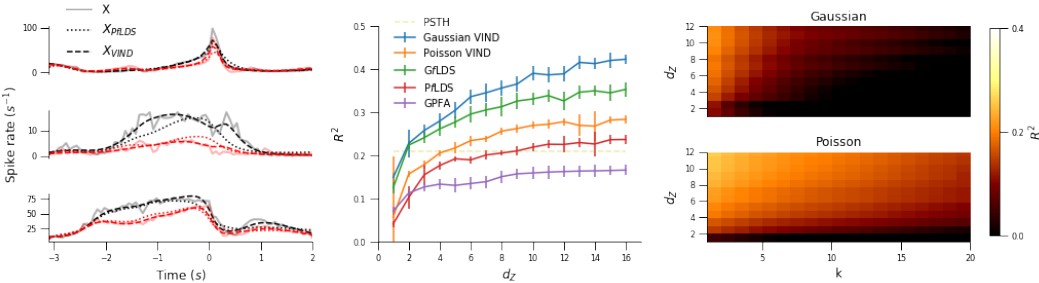

Figure 2: Electrophysiology data. (left) Sample cell spike rates, $t = 0$ signals the start of the response epoch (center) Performance of explained variance ($R^2$) using different setups of VIND and other models. (right) Performance of forward interpolation ($R_k^2$) using two setups of VIND models.

epochs; some neurons show ramping and persistent activity from sample to delay, which relates to the preparation of the choice at response Guo et al. (2014); Li et al. (2015); Wei et al. (2018), while some other neurons show the peaking activity in response to the behavioral epochs, see Fig. (2, left).

We asked whether VIND can capture the variety of neural dynamics using a few latent observations. The data was fitted for $d_Z = 5$, using a Poisson observation model. The fit not only reproduces the neural observation, but also provides insights to the dynamics in the latent space. Details can be found in Fig. 6 in App. B. Subsequently, a 10-fold cross-validation method was used to decide the performance of fit using VIND's Gaussian and Poisson observation models with up to 12 dimensions in the latent space, regardless of trial type. The $R^2$ was computed to determine the performance of VIND as compared to other models. For VIND, both Poisson and Gaussian observation models were used. These are compared to a Peristumulus Time Histogram (PSTH), a GPFA model Yu et al. (2009), as well as GfLDS and PLDS Archer et al. (2015); Gao et al. (2016). The results are shown in the center panel in Fig. 2. We found that nonlinear Gaussian VIND performs the best regarding explained variance of the data.

Next we analyzed the forward interpolation capabilities of the fitted models to the neural data using $R_k^2$. In this case the Poisson observation model gives a substantially better forward interpolation, signaling a dynamical system that more accurately represents the data evolution. This can be seen in the right panel in Fig. 6. These two results combined exemplify the VIND tradeoff between explained variance and forward interpolation capabilities. Using Poisson observations, VIND is less able to fit the higher frequency components of the data. The resulting dynamical system, however, is smoother and more appropriately captures the evolution of the system, see App. B.

### 4.3 SINGLE CELL VOLTAGE DATA

We demonstrate VIND's versatility to uncover underlying dynamics of a system by applying it to 1D voltage electrophysiology data recorded from single cells. This is not a dimensionality reduction problem but rather one of recovering the latent phase space from a single variable to identify the 'true dimensionality' of the system under study. The data is the publicly available Allen Brain Atlas dataset Jones et al. (2009).

Intracellular voltage recordings from cells from the Primary Visual Cortex of the mouse, area layer 4 were selected. Trials with no spikes were removed, resulting in 44 trials from 7 different cells. The input for each of the remaining trials consists of a step-function with an amplitude between 80 and 151pA. Observations were split into training (30 trials) and validation sets (14 trials). The data was then down-sampled from $50,000$ time bins (sample rate of 50 kHz) to $5,000$ in equal-time intervals, and subsequently normalized by dividing each trial by its maximal value.

LLDS/VIND was fit to this data for $d_Z = 2, \ldots, 8$, repeated across 10 runs. The top three fits were averaged and the results are summarized in Fig. 3. The center panel displays the $R_{10}^2$ values for each choice of latent dimensionality. The fits consistently improve up to $d_Z = 5$, after which there are diminishing returns. Single cell voltage data has traditionally been modeled using variants of the classical Hodgkin-Huxley neuron model (Hodgkin & Huxley (1952)), a set of nonlinear differential

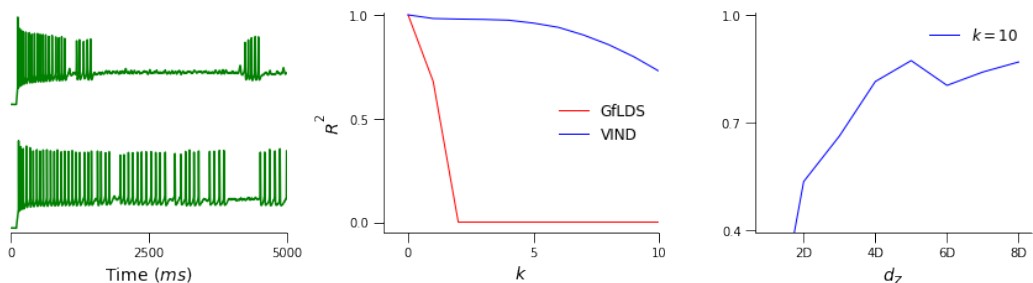

Figure 3: Summary of the LLDS/VIND fit to the Allen dataset: (left) The dataset, neurons respond to an input current; (center) VIND vs GfLDS comparison for the best 5D fits; (right) $R_{10}^2$ for different dimensions. The performance increases up to $d_Z = 5$ possibly indicating the hidden dimensionality of the system.

equations in 4 independent variables, plus an optional independent input current. It is interesting that 5 is exactly the minimal number of latent dimensions that provide a good VIND fit for this data. The right panel displays $R_k^2$ with $d_Z = 5$ for VIND and for GfLDS. VIND outperforms GfLDS by an order of magnitude.

We show the forward-interpolated observations and sample paths for selected runs using VIND and GfLDS in Fig. 4 . The left panel represents the observations over a rolling window, $k = 10$ time-points in advance for both VIND and GfLDS. The dynamics inferred by GfLDS are unable to capture the nonlinearities for both the hyperpolarization and depolarization epochs. The latent trajectories for this data are plotted in the center panel. The dimensions inferred by VIND exhibit similar behavior to that of Hodgkin-Huxley gating variables. As shown in the right panel, we find that in state-space, spikes are represented by big cycles (red), while interspiking fluctuations correspond to separate regions of phase space (blue). For each trial, the height of the voltage spike is coded in the diameter of the cycles.

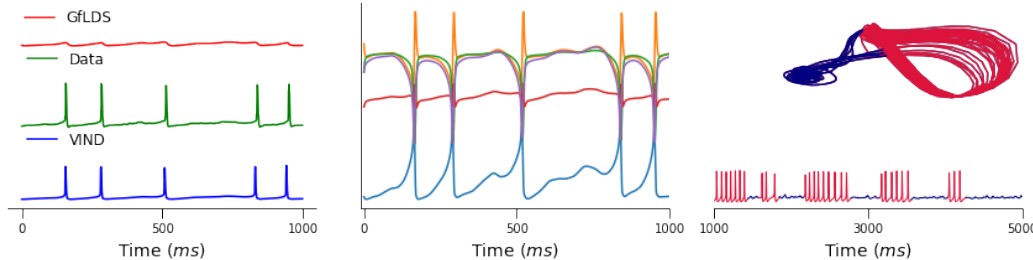

Figure 4: Inferred sample paths: (left) Original data (green) versus the 10-step (2ms) forward interpolation given by VIND and by GfLDS; (center) Latent trajectories for a 5D VIND fit of this data, showing behavior similar to the Hodgkin-Huxley gating variables; (right) A 3D cross-section of the latent space showing the representation of the spikes as big cycles (red) and the transient periods (blue).

### 4.4 SPONTANEOUS ACTIVITY IN WIDEFIELD IMAGING DATA

The unsupervised modeling of spontaneous brain activity is inherently challenging due to the lack of task structure. Here, we model the temporal dynamics of widefield optical mapping (WFOM) data and simultaneous behavior recorded from an awake head-fixed mouse during spontaneous activity Ma et al. (2016a). This data was recorded and corrected for hemodynamics in the Laboratory for Functional Optical Imaging at Columbia University. The recording and preprocessing details for both cortical dynamics and the movement speed signal are provided in App. D. An example frame of the data is shown in Fig. 5 (top-left). The preprocessing of the WFOM cortical data leads to reduced-dimension, denoised cortical activity (as detailed in App. D). The temporal activity of the

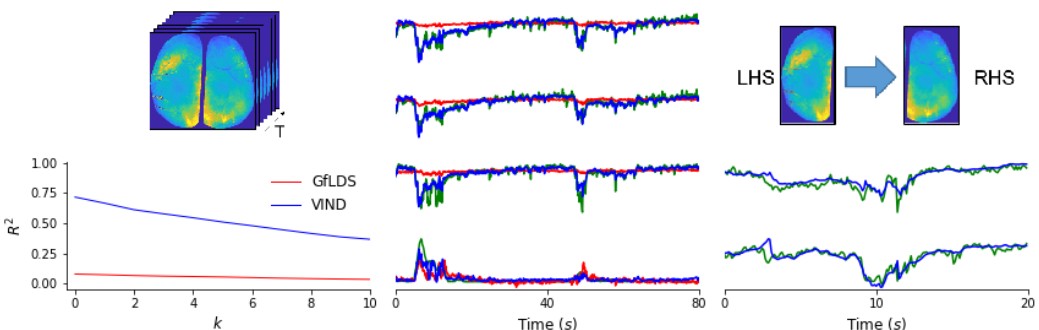

Figure 5: Widefield Imaging Data: (top-left) An example frame of the data. The temporal dynamics and behavior signal are characterized by $X$ after preprocessing, which are simultaneously modeled using both GfLDS and VIND. (bottom-left) Variance weighted average $R^2$ values for $k$-step forward interpolation, with $d_Z = 9$. (center) An example fit of $X$ using VIND on held out data. Only 4 of the signals in the 148-dimensional $X$ signal are shown here. (right) A different VIND model was fit to the temporal dynamics of only the left hand side (LHS) of the brain ($X_{LHS}$). The latents ($Z_{LHS}$) are used to reconstruct the temporal dynamics of the right hand side (RHS) of the video ($X_{RHS}$). Fits are shown on 4 of the 66-dimensional $X_{RHS}$ in held out data.

cortex and the movement speed (jointly called $X$) are simultaneously modeled using both GfLDS and VIND, with the results for validation data on one mouse shown in Fig. 5, where $d_X = 148$, and $d_Z = 9$. The $k-$step forward interpolation is shown in Fig. 5 (bottom-left), with varying $k$, for both VIND and GfLDS. 4 of the 148 dimensions of $X$ and $\hat{X}$ on validation data are shown in Fig. 5 (center). VIND is seen to outperform GfLDS, capturing the fine-tuned dynamics in $X$, thus also leading to better interpolations. We highlight VIND's capability to roughly capture the dynamics of the whole superficial dorsal cortex using a 9-D latent vector and the corresponding evolution and generative network.

Next, a VIND model was fit to the brain dynamics of only the left hand side (LHS) of the brain, after similar preprocessing of the data. Here, $d_{X_{LHS}} = 60, d_{Z_{LHS}} = 9$. A separate neural network was fit from the latents learned on the left hand side ($Z_{LHS}$) to the temporal dynamics of the right hand side (RHS) of the brain ($X_{RHS}$; $d_{X_{RHS}} = 66$), with an MSE loss function. The goal was to infer dynamics from one half of the brain to the other. 5 out of 66 reconstructions of the temporal dynamics of the RHS in held-out data are shown in Fig. 5 (right) (variance weighted average $R^2 = 0.49$ for entire data). For comparison, we ran a baseline CCA analysis which yielded an $R^2$ of 0.45. This shows that the latent variables learned by VIND on one half of the brain are useful to coarsely reconstruct the temporal dynamics of the other half.

## 5    DISCUSSION

In this work we introduced VIND, a novel variational inference framework for nonlinear latent dynamics that is able to handle intractable distributions. We successfully implemented the method for the specific case of Locally Linear Dynamical Systems, which allows for a fast inference algorithm (linear in $T$). When applied to real data, VIND consistently outperforms other methods, in particular methods that rely on an approximate posterior representing linear dynamics. Furthermore, VIND's fits yield insights about the dynamics of these systems. Highlights are the ability to identify the transition points and distinguish among trial types in the electrophysiology task, the dimensionality suggested by VIND's fits for the single-cell voltage data, and the ability of the latents learned from one half of the brain to reconstruct activity from the other half in widefield imaging data. Moreover, VIND can be naturally extended to handle labelled data and data with inputs. This is work in progress.

LLDS/VIND is written in tensorflow and the source code is publicly available.

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

**Algorithm 1** Learning VIND: At every epoch $\mathbf{P}_i^{(\text{ep})}$ is the numerical estimate of the hidden path corresponding to batch $i$, while $\mathbf{P}_{\phi,\varphi}^{(\text{ep})}(\mathbf{X}_i)$ is the $\phi, \varphi$-dependent posterior mean.

---

Initialize $\phi, \varphi, \theta \leftarrow \phi^{(0)} \varphi^{(0)}, \theta^{(0)}$.
**for all** $i$ **do**
    Initialize $\mathbf{P}_i^{(\text{ep})} \leftarrow \mathbf{P}_i^{(0)}$
**end for**
$\text{ep} \leftarrow 1,$
**while** not converged **do**
    **for all** $i$ **do**
        $\mathbf{P}_{\phi,\varphi}^{(\text{ep})}(\mathbf{X}_i) \leftarrow r_{\phi,\varphi}(\mathbf{P}_i^{(\text{ep}-1)}, \mathbf{X}_i).$
        $\mathbf{C}_{\phi,\varphi}^{(\text{ep})}(\mathbf{X}_i) \leftarrow s_{\phi,\varphi}(\mathbf{P}_i^{(\text{ep}-1)}, \mathbf{X}_i)$
        Get $\mathbf{Z}_i \sim q_{\phi,\varphi}(\mathbf{Z}|\mathbf{X}_i) = \mathcal{N}\left(\mathbf{P}_{\phi,\varphi}^{(\text{ep})}(\mathbf{X}_i), \left(\mathbf{C}_{\phi,\varphi}^{(\text{ep})}(\mathbf{X}_i)\right)^{-1}\right)$
    **end for**
    Perform ADAM gradient descent on $\sum_i \mathscr{L}_{\text{ELBO}}(\mathbf{X}_i, \mathbf{Z}_i)$: Update $\phi, \varphi, \theta \leftarrow \phi^{(\text{ep})} \varphi^{(\text{ep})}, \theta^{(\text{ep})}$
    $\mathbf{P}_i^{(\text{ep})} \leftarrow \mathbf{P}_{\phi,\varphi}^{(\text{ep})}(\mathbf{X}_i)|_{\phi^{(\text{ep})} \varphi^{(\text{ep})}}$
    $\text{ep} \leftarrow \text{ep} + 1.$
**end while**

---

# A LLDS/VIND

In this appendix, we provide details of the VIND framework for the LLDS parameterization of the hidden dynamics.

## A.1 ALGORITHM

As detailed in Alg. 1, each training epoch consists of two steps that are carried in alternate fashion: the FPI that updates the best estimate of the latent path and a gradient descent step that updates the model parameters. For optimization of the ELBO we used stochastic gradient descent. As it is customary, in order to estimate the gradients, the so called "reparameterization trick" is used. Samples are extracted from the variational approximation $q_{\phi,\varphi}$:

$$\mathbf{Z}_i = \mathbf{P}_{\phi,\varphi}^{(\text{ep})}(\mathbf{X}_i) + \left[\mathbf{C}_{\phi,\varphi}^{(\text{ep})}(\mathbf{X}_i)\right]^{-1/2} \epsilon \tag{23}$$

where $\epsilon$ is a standard normal, (Kingma & Welling (2013); Jimenez Rezende et al. (2014)).

## A.2 IMPLEMENTATION DETAILS

We provide here extra details of the LLDS/VIND implementation that was used throughout the paper:

- The initial path estimates $\mathbf{P}_i^{(0)}$ are taken to be $\boldsymbol{\mu}_\varphi(\mathbf{X}_i)$.
- VIND is initialization-sensitive. Empirically, we found that it is important that the initial path estimates fall within a region where the nonlinearity is not severe ($\max_{\mathbf{P}_i} |A_\phi(\mathbf{z}_t) - \mathbb{I}| \lesssim 0.1$ for every trial $i$).
- To encourage smoothness of the latent dynamics, $A_\phi(\mathbf{z}_t)$ was specified as

$$A_\phi(\mathbf{z}_t) = \mathbb{A} + \alpha \cdot B_\phi(\mathbf{z}_t) \tag{24}$$

  where $\mathbb{A}$ is a state-space-independent linear transformation initialized to the identity, $\alpha$ is a nontrainable hyperparameter of the model, and $B_\phi(\mathbf{z}_t) = \text{NN}_{\phi_B}(\mathbf{z}_t)$. This set up has the added benefit that $\alpha = 0$ is equivalent, both the statistical model and the algorithm, to GfLDS/PfLDS (Archer et al. (2015); Gao et al. (2016)).
- The local transformation $A_\phi(\mathbf{z}_t)$ is redundant (it is akin to a gauge transformation in physics parlance). To see this, note that for every $\mathbf{z}_t$, the image of the transformation $A_\phi(\mathbf{z}_t)\mathbf{z}_t$ is at most $\mathbb{R}^n$. On the other hand $A_\phi(\mathbf{z}_t)$ has dimensionality $\mathbb{R}^{n^2}$. In other words, given $\mathbf{z}_t$ and

$\mathbf{z}_{t+1}$, there is a continuum of matrices $A_\phi(\mathbf{z}_t)$ that satisfy $\mathbf{z}_{t+1} = A_\phi(\mathbf{z}_t)\mathbf{z}_t$. It follows that $A_\phi(\mathbf{z}_t)$ can be substantially restricted without loss of generality ("fixing the gauge"). We found that the best results were obtained when $A_\phi(\mathbf{z}_t)$ was constrained to be symmetric.

- Experiments showed that setting the number of FPIs at $n = 2$ in Eq. (20) is enough for producing good convergence results across datasets.

- In all experiments we found that there is no noticeable decrease in performance if the gradient terms in Eq. (19), and the corresponding ones for $s_{\phi,\varphi}$ are neglected. These terms are subleading compared to $\mathbf{\Lambda}_\varphi \mathbf{M}_\varphi$ both because they are proportional to the nonlinearity, small as required by smoothness, and because the gradient is applied on a deep neural network.

- The FPI in Eq. (12) is in the contractive regime within a domain $D$, $\mathbf{P}_i^{(0)} \in D$; $D \subset \mathbb{R}^{T \times d_z}$ when the Jacobian $\mathbf{J}$ of the map $r_{\phi,\varphi}$ satisfies that the absolute value of its determinant is smaller than 1. This can be guaranteed for any starting point $\mathbf{P}_i^{(0)}$ by invoking the Gershgorin Circle Theorem (Eisstein), which yields a bound on the absolute value of the eigenvalues of a matrix. For the specific case of LLDS/VIND, the entries of $\mathbf{J}$ are suppressed both by the small hyperparameter $\alpha$ and to the gradients of the deep neural network $B_\phi(\mathbf{z}_t)$, Eq. (24. This makes the Gershgorin bound easy to satisfy. Convergence of the algorithm is not straightforward from this argument, since it must cycle through different batches of training data, which may lie in different basins of the FPI map. Nevertheless, in practice we found the Jacobian suppression to be enough to guarantee convergence.

# B   DETAILS OF THE ELECTROPHYSIOLOGY FIT

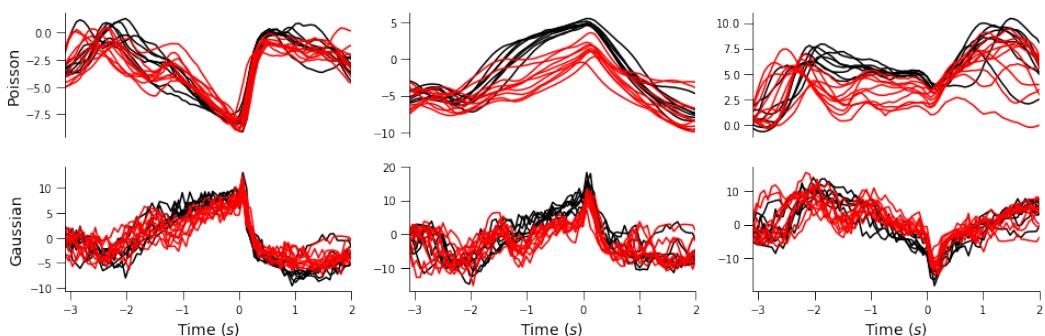

Figure 6: Examples of latent dimension dynamics for Gaussian and Poisson VIND in validation data. Black lines, posterior pole location; red lines, anterior pole location. Notice how the inferred paths differ for posterior and anterior pole locations. Also note visible changes in dynamics at $t = -1.3$ (stimulus), and $t = 0$ (go cue).

The experimental details in Sec. 4.2 are as follows. The recording session contains 18 simultaneous recorded units, with 74 lick-right trials (in blue; posterior pole location) and 100 lick-right trials (in red; anterior pole location). Spike counts were binned in a 67 ms non-overlapped time window, where the maximum spike count is 10 and minimum is 0. The fit covers time points (almost the entire recordings) from -0.5 sec from sample onset to 2.0 sec from response (wherefore the trial contains 77 time bins).

Fig. 6A shows the average neuronal activity of 3 representative cells in the recordings. Cell #1 is a typical neuron with small separation of trials, but strong peaking activity at transition from delay to response epochs. Cells #2, 3 exhibit the stereotypical ramping activity and separations of different trial types, which are assumed for preparation of the movements. Both setups of VIND models (Poisson and Gaussian, nonlinear evolutions, $d_z = 5$) can reproduce the complex and variable neural dynamics in the held-out trials (9 lick-left trials; 9 lick-right trials). Particular, Gaussian VIND model can capture the changes of dynamics on even finer timescales (dash lines vs dot lines; Poisson vs Gaussian VIND models). This observation agrees with higher performance of explained variance ($R^2$; Fig. 2).

The latent dynamics are smoother in the Poisson VIND model, (Fig. 6BC). We have found that in VIND fits, smoother trajectories are correlated with superior performance in the forward interpolation tasks. Intuitively, for noisier latent paths, the algorithm attempts to ascribe some of the variance to the dynamical system, which hurts the forward interpolation capabilities. In the Poisson VIND fit represented in Fig. 6, the latent dynamics in dimension 2 and 3 appears to represent the preparation of the choice where the neural dynamics for different trial types gradually diverges with time. The dynamics in dimension 1 shows rapid peaking dynamics at the transitions of the behavioral epochs. However, those two types of dynamics were mixed and separations of trial types were in Gaussian VIND model. In general, ramping and peaking dynamics is not operated by distinguishable groups of neurons, yet to our surprise they are separated in the latent space.

## C DETAILS OF THE SINGLE CELL VOLTAGE DATA FITS

For the Allen data, Fig. 7 shows simulated paths (forward interpolation with noise) versus the corresponding real data. Fig. 8 shows several views of the same two latent paths corresponding to two different input currents.

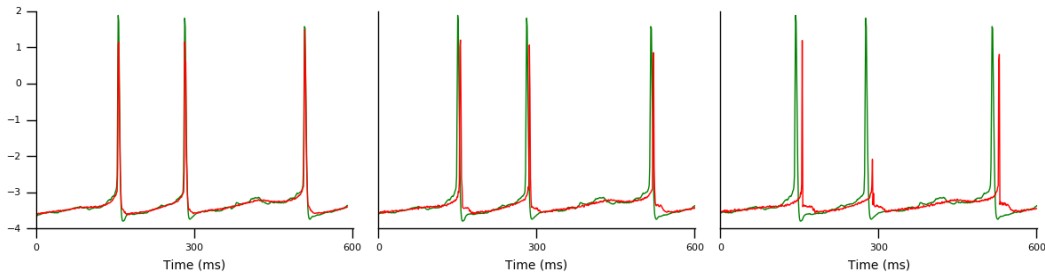

Figure 7: Data (green) versus simulation of the observations (red) from the smoothed path: 10 steps ahead (left), 20 steps ahead (center), and 30 steps ahead (right). Some signs of deterioration of the prediction start to appear for the latter (failed spikes, late spiking times).

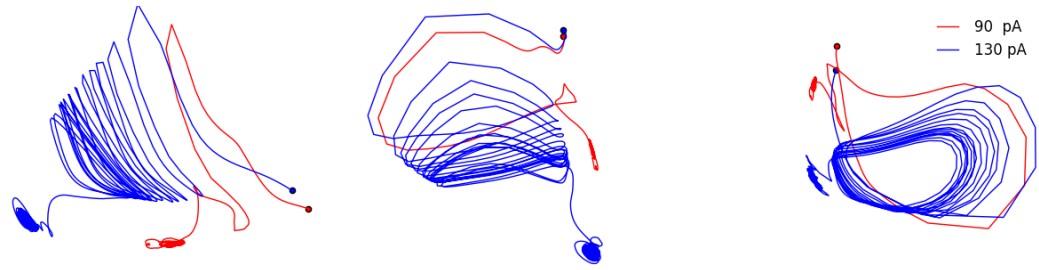

Figure 8: Different views of a 3D cross section of 5D latent paths for two different trials, showing how the paths occupy different regions of state-space depending on the value of the constant input current.

## D PREPROCESSING OF WIDEFIELD IMAGING DATA

Macro-scale wide-field optical mapping (WFOM) is an increasingly popular technique for surveying neural activity over very large areas of cortex with high temporal resolution. WFOM can image the fluorescence of genetically-encoded calcium (GCaMP6f) indicators using LED illumination and camera detection scheme. We use methods for correcting fluorescence recordings of neural activity for confounding contamination by changes in hemoglobin concentration and oxygenation as in Ma et al. (2016b), by measuring both neural fluorescence signals and hemodynamics. This correction provides us with an accurate change in fluorescence of neural regions ($\Delta F/F$). An example frame

of the data is shown in Fig. 5A, 464-by-473 pixels. The activity of the mouse is simultaneously recorded using a webcam pointed at the mouse's body, and the movement speed at time $t$ is taken as a 1D signal consisting of the standard deviation of the difference in value of all pixels from time $t - 1$ to time $t$.

We use a WFOM recording of length 2 minutes, where the signals are sampled at 10Hz, thus leading to 1200 time points. We normalize $\Delta F/F$ to lie between 0 and 1 for every video, and then apply block singular value decomposition (SVD) to the videos for denoising and dimensionality reduction Buchanan et al. (2018). First, we fit an anisotropic Wiener filter in a $4 \times 4$ neighborhood of each pixel to reduce uncorrelated noise while preserving spatially-local, time-correlated signals. Next, the video is partitioned into 25 ($5 \times 5$) blocks, and SVD is performed on the pixels in each block. The temporal components are ranked according to a metric defined on their empirical autocorrelation function, and components that fall within a 99% confidence interval of Gaussian white noise are discarded. Moreover, those temporal components that have a signal-to-noise ratio lower than 1.6 are also discarded. The remaining temporal components from each block are concatenated, and these form the $X$ matrix, here $147 \times 1200$. This is augmented using a 1D behavior signal that is extracted using the standard deviation of successive frames from a webcam recording the lateral view of the mouse's body, representing the speed of the mouse's movements in arbitrary units. We used different sessions of recording from the same mouse, preprocessed in the same way, to obtain training and validation data.

