# OpenReview forum: "A NOVEL VARIATIONAL FAMILY FOR HIDDEN NON-LINEAR MARKOV MODELS"
_ICLR.cc/2019/Conference_

### Official Review · AnonReviewer3 · 2018-10-29
**Incremental technical contribution but with extensive experimental evaluation**

**Rating:** 6
**Confidence:** 3

**Review:**

The paper presents a variational inference approach for locally linear dynamical models. In particular,  the latent dynamics are drawn from a Gaussian approximation of the parent variational distribution,  enabled by Laplace approximations with fixed point updates, while the parameters are optimized the resulting stochastic ELBO. Experiments demonstrate the ability of the proposed approach to learning nonlinear dynamics, explaining data variability, forecasting and inferring latent dimensions.

Quality: The experiments appear to be well designed and support the main claims of the paper.

Clarity: The clarity is below average. In Section 2 the main method is introduced. However, the motivation and benefits of introducing a parent and child variational approximation are not discussed adequately. It would be helpful to move some of the stuff in the appendix to the main text, and present in a neat way. I also struggled a little to understand what is the difference between forward interpolate and filtering.

Originality: Given the existing body of literature, I found the technical novelty of this paper rather weak. However, it seems the experiments are thoroughly conducted. In the tasks considered, the proposed method demonstrates convincing advantages over its competitors.

Significance: The method shall be applicable to a wide variety of sequential data with nonlinear dynamics.

Overall, this appears to be a board-line paper with weak novelty. On the positive side, the experimental validation seems well done. The clarity of this paper needs to be strengthened.

Minor comments:
- abstract: uncover nonlinear observation? -> maybe change "observation" to "latent dynamics"?

---

> ### Author Response · Authors · 2018-11-26
> **Response to AnonReviewer3**
>
> We thank the reviewer for the useful comments, below our replies.
>
> 1. "The clarity is below average. In Section 2 the main method is introduced. However, the motivation and benefits of introducing a parent and child variational approximation are not discussed adequately. It would be helpful to move some of the stuff in the appendix to the main text, and present in a neat way."
>
> We fully agree with this criticism. Following your suggestion (and also that of reviewer # 2) we have moved some material from the appendix to the main text.
>
> 2. "I also struggled a little to understand what is the difference between forward interpolate and filtering"
>
> In this work we refer by filtering to the process of inferring the optimal latent state z_t  at time t, using observations x_{1:t} from the trial up to time t, not including observations to the future of t. By forward interpolation we refer to the process of smoothing, (inferring optimal z_t from observations of the complete trial x_{1:T}, including points to the future of t), and then evolving the inferred z_t with the learned VIND dynamics. After evolving for k steps, the Generative Model is used to generate data which is subsequently compared with the observations at time t+k. We do not refer to this procedure as “prediction” since the initial state z_t for the forward interpolation was obtained by making use of the full data.
>
> We have added clarifying comments at the beginning of section 4.
>
> 3. "Given the existing body of literature, I found the technical novelty of this paper rather weak"
>
> We would like to reiterate that the novelty of the paper is <i>twofold.</i>
>
> First and foremost, we propose the use of a novel variational approximate posterior that shares the nonlinear dynamics with the generative model. This feature is powerful because it uses known information about the true posterior in the design of the approximate one. Naively, the feature also seems to be a curse because the variational approximation is rendered intractable for the case of nonlinear dynamics. This is the reason why such approximate posteriors have not been proposed before. We have added a sentence in the introduction emphasizing this crucial point.
>
> The second novelty is a method to deal with this intractability, via the Laplace approximation and the fixed-point iteration method. We showed that the resulting algorithm, which intercalates a gradient step and a FPI step yields very good results in well-known, difficult tasks such as dimensionality expansion in the single cell data or the WFOM task.
>
> 4.- "abstract: uncover nonlinear observation? -> maybe change "observation" to "latent dynamics"?"
>
> The term ‘nonlinear observation’ in the line “…Variational Inference for Nonlinear Dynamics (VIND), that is able to uncover nonlinear observation and transition functions from sequential data …“, found in the abstract refers to the observation map in the Generative Model. That is, VIND uncovers both a nonlinear “observation” model, that maps nonlinearly a latent state to the data, and nonlinear latent dynamics mapping the latent state at time t to the state at time t+1, which we refer to as the “nonlinear transition functions”.
>
> On the other hand, we agree that “nonlinear latent dynamics” is a better fit than “transition functions” for the abstract and we have performed this replacement.

---

### Official Review · AnonReviewer1 · 2018-11-03
**Excellent method and results but need more comparisons and better writing**

**Rating:** 8
**Confidence:** 5

**Review:**

I'll start with a disclaimer: I have reviewed the NIPS 2019 submission of this paper which was eventually rejected. Compared to the NIPS version, this manuscript had significantly improved in its completeness. However, the writing still can be improved for rigor, consistency, typos, completeness, and readability.

Authors propose a novel variational inference method for a locally linear latent dynamical system. The key innovation is in using a structured "parent distribution" that can share the nonlinear dynamics operator in the generative model making it more powerful compared. However, this parent distribution is not usable, since it's an intractable variational posterior. Normally, this will prevent variational inference, but the authors take another step by using Laplace approximation to build a "child distribution" with a multivariate gaussian form. During the inference, the child distribution is used, but the parameters of the parent distribution can still be updated through the entropy term in the stochastic ELBO and the Laplace approximation. They use a clever trick to formulate the usual optimization in the Laplace approximation as a fixed point update rule and take one fixed point update per ADAM gradient step on the ELBO. This allows the gradient to flow through the Laplace approximation.

Some of the results are very impressive, and some are harder to evaluate due to lack of proper comparison. For all examples, the forward interpolate (really forecasting with smoothed initial condition) provides a lot of information. However, it would be nice to see actual simulations from the learned LLDS for a longer period of time. For example, is the shape of the action potential accurate in the single cell example? (it should be since the 2 ms predictive r^2 shows around 80%).

Except in Fig 2, the 3 other examples are only compared against GfLDS. Since GfLDS involves nonconvex optimization, it would be reasonable to also request a simple LDS as a baseline to make sure it's not an issue of GfLDS fitting.

For the r^2=0.49 claim on the left to right brain prediction, how does a baseline FA or CCA model perform?

Was input current ignored in the single cell voltage data? Or you somehow included the input current as observation model?

As for the comment on Gaussian VIND performing better on explaining variance of the data even though it was actually count data, I think this maybe because you are measuring squared error. If you measured point process likelihood or pseudo-r^2 instead, Poisson VIND may outperform. Both your forecasting and the supplementary results figure show that Poisson VIND is definitely doing much better! (What was the sampling rate of the Guo et al data?)

The supplementary material is essential for this paper. The main text is not sufficient to understand the method.

This method relies on the fixed point update rule operating in a contractive regime. Authors mention in the appendix that this can be *guaranteed* throughout training by appropriate choices of hyperparameters and network architecture. This seems to be a crucial detail but is not described!!! Please add this information.

There's a trial index suddenly appearing in Algorithm 1 that is not mentioned anywhere else.

Is the ADAM gradient descent in Algorithm 1 just one step or multiple?

MSE -> MSE_k in eq 13

LFADS transition function is not deterministic. (page 4)

log Q_{phi,varphi} is quadratic in Z for the LLDS case. Text shouldn't be 'includes terms quadratic in Z' (misleading).

regular gradient ascent update --> need reference (page 4)

Due to the laplace approximation step, you don't need to infer the normalization term of the parent distribution. This is not described in the methods (page 3).

Eq 4 and 5 are inconsistent in notation.

Eq (1-6) are not novel but text suggests that it is.

Predict*ive* mean square error (page 2)

Introduction can use some rewriting.

arXiv papers need better citation formatting.

---

> ### Author Response · Authors · 2018-11-26
> **Response to AnonReviewer2**
>
> We thank the reviewer for the comprehensive review. We respond below to all the  arguments/objections.
>
> 1.- "...it would be nice to see actual simulations from the learned LLDS for a longer period of time. For example, is the shape of the action potential accurate in the single cell example? (it should be since the 2 ms predictive r^2 shows around 80%)."
>
> As you recommend, we have included a new appendix with more figures on the Allen data fits. In particular we show a simulation from the learned LLDS for 10, 20 and 30 time steps ahead. In particular, at 30 steps the dynamics is still producing spikes at roughly the right times although some deterioration of performance becomes evident.
>
> 2.- "Except in Fig 2, the 3 other examples are only compared against GfLDS. Since GfLDS involves nonconvex optimization, it would be reasonable to also request a simple LDS as a baseline to make sure it's not an issue of GfLDS fitting."
>
> We tried fitting an LDS to the Lorenz data using the PyKalman library that learns the LDS using a standard implementation of the EM algorithm. The algorithm, applied to a dataset of multiple trials with Gaussian observations, was unable to converge for a 3D latent space.The algorithm does converge for a dataset consisting of a single trial but the found dynamics is not generalizable. More than that, the single-trial dynamics performs comparatively poorly even when tested in the data used for training, with the 1-step forward interpolation on training data yielding an average k=1 R2 of .938 (compare with VIND’s k=1 R^2 of .998 on the Lorenz validation data)
>
> The EM algorithm for learning the LDS does not yield meaningful results in the case of dimensionality expansion (Allen data) either. In that case, it simply copies the data to one of the latent space dimensions and yields the identity transition function. For all these reasons we thought that the LDS baseline was not very informative and decided not to include.
>
> 3.- "For the r^2=0.49 claim on the left to right brain prediction, how does a baseline FA or CCA model perform?"
>
> Following your suggestion, we performed a baseline CCA analysis. It provided an R^2 of 0.45, which is smaller, though comparable to the R^2 of 0.49 of VIND. We have included this in the manuscript. Of course, on top of the superior R^2, VIND also has the power to provide a prediction, since it fits a dynamical model, which CCA is not capable of providing.
>
> 4.- "Was input current ignored in the single cell voltage data? Or you somehow included the input current as observation model?"
>
> The data used in the single-cell voltage experiments was taken from samples were the input current was held constant throughout each trial (although not across different trials). Therefore, for this dataset, the input current behaves as a parameter that varies per trial and roughly determines the region of phase space occupied by the latent trajectory.
>
> In the new appendix we have included a plot where the latent paths of two trials, corresponding to different input currents, are shown. The plot illustrates how these trials occupy different regions in the latent state which we interpret, at least partly, as the representation of the constant input current as a coordinate in the latent space. We should further say that although it is not included in this manuscript, we are working in an extension of VIND to find latent dynamics that accepts arbitrary inputs, such as time-varying current in the Allen data.

---

> ### Author Response · Authors · 2018-11-26
> **Response II to AnonReviewer2**
>
> 5.- "... Gaussian VIND performing better ... pseudo-r^2 instead, Poisson VIND may outperform...."
>
> The sample rate of the data was 60 kHz ephys and binned to 67 ms. We have added this information in the text.
>
> We agree that Poisson VIND is performing much better, especially at forecasting, and also yields smoother dynamics. We are not sure however about how to perform a meaningful comparison of the model with Gaussian observations and the model with Poisson observations using a pseudo R^2. These two models have different likelihoods so - as opposed to the regular R^2 that we used - typical pseudo R^2s, like McFadden’s, would be computing different quantities.
>
> 6.- "The supplementary material is essential for this paper. The main text is not sufficient to understand the method."
>
> Yes. When designing the paper, and due to the length constraints, we faced a decision between writing a more theoretical paper or writing a paper emphasizing the usefulness of VIND in a varied set of tasks. We ultimately decided for the latter which resulted in important information on the methods been presented as supplementary material.
>
> Following your suggestion (and that of reviewer #3) we have moved portions of the theoretical appendix to the main text.
>
> 7.- "This method relies on the fixed point update rule operating in a contractive regime. ... Please add this information."
>
> Yes! The fixed-point iteration is in the contractive regime when the absolute value of the determinant of the Jacobian of the iterative map (r in Eq. 12) is smaller than 1. This can be guaranteed for example by ensuring that the entries of the Jacobian are small enough and then invoking the Gershgorin Circle Theorem. For LLDS/VIND, the Jacobian of r is proportional to both the hyperparameter alpha and to the gradients of the evolution network with respect to the latent state. These two are indeed required to be relatively small in order  to guarantee the smoothness of the evolution. For instance, in our experiments, we found that choosing alpha ~ 10^-1, and a softplus nonlinearity in the next-to-last hidden layer of the evolution network, ensured gradients small enough to be in the contractive regime as desired.
>
> We have added a paragraph in Appendix A addressing this point.
>
> 8.- "There's a trial index suddenly appearing in Algorithm 1 that is not mentioned anywhere else."
>
> We meant a batch index. We fixed it in the text.
>
> 9.- "Is the ADAM gradient descent in Algorithm 1 just one step or multiple?"
>
> We kindly ask the reviewer to clarify whether this question refers to a) updating all the trainable parameters at once versus specific subsets in some order or b) performing multiple gradient descent steps per FPI within one epoch. If a) then, it is a one step ADAM gradient descent. Regarding b),we tried different setups. One gradient descent step per FPI appears to yield the best results.
>
> 10.- "MSE -> MSE_k in eq 13"
>
> Fixed!
>
> 11.- "LFADS transition function is not deterministic. (page 4)"
>
> We agree that the sentence was ambiguous.
>
> In order to compare VIND to LFADS (or to any other model) we would argue that the fair comparison is among the respective Generative Models. In our understanding, LFADS GM, as read for instance in Eqs. 1-6 in arXiv:1608.06315 has a deterministic transition function, Eq. (3), with a stochastic input. We agree however that the evolution of the full LFADS model is not deterministic due to the presence of the back link from the GM factors to the controller. This turns the evolution of full LFADS, Generative plus Recognition, non-deterministic.
>
> We have removed that sentence from the manuscript
>
> 12.- "log Q_{phi,varphi} is quadratic in Z for the LLDS case. Text shouldn't be 'includes terms quadratic in Z' (misleading)."
>
> But our log Q_{phi,varphi} is not strictly quadratic in Z for the LLDS, right?, since it contains A(Z) which is an arbitrary nonlinearity?
>
> 13.- "regular gradient ascent update --> need reference (page 4)"
>
> Fixed!
>
> 14- "Due to the laplace approximation step, you don't need to infer the normalization term of the parent distribution. This is not described in the methods (page 3)."
>
> Indeed! Added clarifying sentence.
>
> 15.- "Eq 4 and 5 are inconsistent in notation."
>
> Fixed!
>
> 16.- "Eq (1-6) are not novel but text suggests that it is."
>
> We improved the text around Eqs. (1-6) and added citations to eliminate misleading claims.
>
> 17.- "Predict*ive* mean square error (page 2)"
>
> Fixed.
>
> 18 - "arXiv papers need better citation formatting."
>
> We have fixed the arXiv citation style to include arXiv preprint numbers (they seem to be removed by default by the ICLR style file?)

---

### Official Review · AnonReviewer2 · 2018-11-03
**Nice algorithm but need better motivation**

**Rating:** 5
**Confidence:** 3

**Review:**

This paper discusses a algorithm for variational inference of a non-linear dynamical models. In this paper model assumption is to use single stage Markov model in latent space with every latent variable Z_t to be defined Gaussian distributed with mean depends on Z_(t-1) and time invariant variance matrix lambda. The non linearity in transition is encoded in mean of Guassian distribution. For modeling the likelihood and observation model, the Poisson or Normal distribution are used with X_t being sampled from another Gaussian or Poisson distribution with the non-linearty being encoded in the parameters of distribution with variable Z_t.  This way of modeling resembles so of many linear dynamical model with the difference of transition and observation distribution have nonlinearity term encoded in them.
The contribution of this paper can be summarized over following points:

- The authors proposed the nonlinear transition and observation model and introduced a tractable inference model using Laplace approximation in which for every given set of model parameter solves for parameters of Laplace approximation of posteriori and then model parameters get updated until converges

-the second point is to show how this model is successful to capture the non-linearity of the data while other linear models do not have that capabilities


Novelty and Quality:
The main contribution of this paper is summarized above. The paper do not contain any significant theorem or mathematical claims, except derivation steps for finding Laplace approximation of the posteriori. The main challenge here is to address effectiveness of this model in comparison to other non-linear dynamical system that we can name papers as early as Ghahramani, Zoubin, and Sam T. Roweis. "Learning nonlinear dynamical systems using an EM algorithm." Advances in neural information processing systems. 1999.
or more recent RNN paper LSTM based papers. I think authors need to distinguish what this paper can give to community beside approximate posteriori of latent variables that other competing models are not capable of. If the aim is to have that posteriori, the authors should show what type of interpretation they have drawn from that in experiments.
There are lots of literature exist on speech, language models and visual prediction which can be used as reference as well.

Clarity:
The paper is well written and some previous relevant methods have been reviewed . There are a few issues that are listed below:

1- as mentioned in Quality sections authors should be more clear about what is distinguished in this paper that other non-linear dynamical systems

2- they used short form RM for Recognition model or FPI for fixed point iteration that need need to be defined before being used



significance and experiments:
The experiments are extensive and authors have compared their algorithm with some other linear dynamical systems (LDS) competing algorithms and showed improvement in many of the cases for trajectory reconstruction.
A few points can be addressed better, it can be seen for many of experiments exhaustive search is used for finding dimension of latent variable. This issue is addressed in Kalantari, Rahi, Joydeep Ghosh, and Mingyuan Zhou. "Nonparametric Bayesian sparse graph linear dynamical systems." arXiv preprint arXiv:1802.07434 (2018). That paper can use non-parametric approaches to find best latent dimension, although the paper applied the technique on linear system, same technique could be adopted to non-linear models. Also that model is capable of finding multiple linear system that model the non linearity by switching between diffrent linear system, for switching linear system, this paper can be named as well: Linderman, Scott, et al. "Bayesian learning and inference in recurrent switching linear dynamical systems." Artificial Intelligence and Statistics. 2017.

It is shown that the model can reconstruct the spikes very well while linear model do not have that power (which is expected), but it is interesting to see how other non-linear models would compare to this model under those certain conditions

It is desired and interesting to see how the model behave one step ahead and K-step ahead prediction. Please address why it cannot be done if there is difficulties in that.

---

> ### Author Response · Authors · 2018-11-26
> **Response to AnonReviewer1**
>
> We thank the reviewer for the detailed criticism. Below our replies
>
> 1.- "The main challenge here is to address effectiveness of this model in comparison to other non-linear dynamical system that we can name ...  I think authors need to distinguish what this paper can give to community beside approximate posteriori of latent variables that other competing models are not capable of."
>
> We would like to take the chance to emphasize the main contributions of our paper. As the reviewer remarks, one of the contributions of VIND is a novel structured approximate posterior. However, it is a cardinal point of our work that this approximate posterior (the parent distribution in the manuscript) inherits the terms that describe the evolution in the latent space directly from the Generative Model (GM). Thus, more specifically, regarding the terms describing the latent space dynamics, the posterior is in fact not approximate. That derivation is exact, see Eq. 6. This prescription is also what makes the parent distribution intractable by standard methods. Thus, a third contribution of VIND is a novel algorithm, that allows dealing with that intractable posterior; using the Laplace approximation coupled with a Fixed Point Iteration (FPI) step.
>
> To reiterate, to our knowledge, our method is the first one that allows variational inference on an approximate posterior that inherits the exact nonlinear evolution law from the GM. This is a key contribution of our work. We have updated the introduction to clarify this point.
>
> We agree with the reviewer that a comparison with existing methods for training models with nonlinear dynamics is an important aspect. As mentioned in the paper, we considered recent work on Bayesian learning methods such as Deep Kalman Filters (Krishnan et al, 2015), as well as RNN-based methods such as LFADS (Sussillo et al, 2016), for comparison, and provided qualitative differences between these methods and our approach. We struggled to make definitive quantitative comparisons by applying these methods on our data; since (1) both methods required specific tuning of hyperparameters, and there are many in both methods; (2)the k-steps ahead R^2 measure we use for the most part is not easy to obtain from the publicly available code. We are of the opinion that this metric is a more informative measure of how well the model is performing, as compared to the simple R^2 measures. We will continue to pursue making these quantitative comparisons but at this moment, it is still work in progress.
>
> 2.- If the aim is to have that posteriori, the authors should show what type of interpretation they have drawn from that in experiments.
>
> We are unsure about what the reviewer is asking us in this question. In the manuscript we presented several conclusions extracted from VIND’s experiments. In particular, for the Lorenz and single-cell systems we argued that VIND is able to uncover both the underlying dimensions of the system and its dynamics. In the electrophysiology task, we showed that VIND is able to separate between the two trial types corresponding to anterior-posterior pole discrimination. We would like to kindly ask the reviewer to clarify the question.
>
> 3.- as mentioned in Quality sections authors should be more clear about what is distinguished in this paper that other non-linear dynamical systems
>
> As we emphasized above, a crucial point of the paper - that, in particular, makes it different from other methods for variational inference of nonlinear dynamics - is that inference is performed on an evolution law that is read directly from the proposed Generative Model. We believe that this is partly the reason why the evolution of the system (forward-interpolation) with the VIND-trained dynamics performs so well across tasks. We have added sentences in the introduction and the Conclusions to further make this point clear.
>
> We tried our best to make comparisons as mentioned above. Apart from the issues already mentioned, we felt that fair direct comparisons to the methods for inference of nonlinear dynamics mentioned above (LFADS, DKF) were made difficult by the fact that as far as we are aware, they have been mainly tested by their own developers.
>
> 4.- they used short form RM for Recognition model or FPI for fixed point iteration that need need to be defined before being used
>
> We would like to point out that RM for Recognition Model was defined on page 2, right before Eq. 1, and FPI for fixed-point-iteration was defined in the Introduction section to the paper. We believe these are the first instances that these two terms were mentioned.

---

> ### Author Response · Authors · 2018-11-26
> **Response II to AnonReviewer1**
>
> 5.- "...exhaustive search is used for finding dimension of latent variable. ...  non-parametric approaches to find best latent dimension, .... same technique could be adopted ....."
>
> This is a very interesting idea. In this paper the datasets considered were small enough that performing a simple exhaustive search was feasible and we were able to thoroughly explore how the forward-interpolated paths changed as the latent dimensionality increased. We agree with the reviewer that on larger datasets, it would certainly be interesting to apply these methods with VIND in order to determine the best latent dimension.
>
> 6.- "...this paper can be named as well: Linderman, Scott, et al. "Bayesian learning and inference in recurrent switching linear dynamical systems." Artificial Intelligence and Statistics. 2017...."
>
> We were aware of the work by Linderman, cited in the introduction. We have now included a citation to the work of Rahi.
>
> 7.- It is desired and interesting to see how the model behave one step ahead and K-step ahead prediction. Please address why it cannot be done if there is difficulties in that.
>
> In the manuscript we did evaluate all the tasks using what we called the k-step ahead "forward-interpolation". However, this is essentially prediction, but starting from the most accurate possible estimate of the initial point. This criterium is designed to ascertain how well the fitted dynamics can reproduce the known evolution of the data. We only refrained from calling the procedure “prediction” because the whole data is used to estimate the starting point (smoothing). But we do want to emphasize that the only way to determine whether the trained dynamics are a good description of the evolution of the system is to compare synthetic data generated with it with real data. To perform this comparison, we must ensure that the initial latent state, is the best possible estimate of the latent state corresponding to the actual data. This is why that initial smoothing is important.
>
> Although pure prediction is useful for some applications, forward-interpolation is more appropriate for establishing the quality of the learned model.

---

### Meta-Review · Area_Chair1 · 2018-12-14
**borderline - but leaning to reject because of reviewer reservations**

**Confidence:** 4
**Recommendation:** Reject

**Metareview:**

The reviewers in general like the paper but has serous reservations regarding relation to other work (novelty) and clarity of presentation. Given non-linear state space models is a crowded field it is perhaps better that these points are dealt with first and then submitted elsewhere.